TECHNICAL RELEASE

# SMARTER-database: a tool to integrate SNP array datasets for sheep and goat breeds

Paolo Cozzi[1,*], Arianna Manunza[1], Johanna Ramirez-Diaz[1],
Valentina Tsartsianidou[2,3], Konstantinos Gkagkavouzis[2,3], Pablo Peraza[4],
Anna Maria Johansson[5], Juan José Arranz[6], Fernando Freire[7],
Szilvia Kusza[8], Filippo Biscarini[1], Lucy Peters[9], Gwenola Tosser-Klopp[9],
Gabriel Ciappesoni[4], Alexandros Triantafyllidis[2,3], Rachel Rupp[9],
Bertrand Servin[9] and Alessandra Stella[1]

1 Institute of Agricultural Biology and Biotechnology, National Research Council, Via Alfonso Corti nr. 12, 20133, Milano, Italy
2 Department of Genetics, Development & Molecular Biology, School of Biology, Aristotle University of Thessaloniki, 541 24, Thessaloniki, Greece
3 Genomics and Epigenomics Translational Research (GENeTres), Center for Interdisciplinary Research and Innovation (CIRI-AUTH), Thessalonikis-Thermis, 57001, Greece
4 Sistema Ganadero Extensivo, Instituto Nacional de Investigación Agropecuaria, INIA Las Brujas, CP 90200, Uruguay
5 Department of Animal Breeding and Genetics, Swedish University of Agricultural Sciences, 75007, Uppsala, Sweden
6 Departamento de Producción Animal, Facultad de Veterinaria, Universidad de León, 24071, León, Spain
7 OVIGEN, Zamora, Spain
8 Centre for Agricultural Genomics and Biotechnology, University of Debrecen, Debrecen, 4032, Hungary
9 GenPhySE, Université de Toulouse, INRAE, ENVT, 31326, Castanet-Tolosan, France

**Submitted:** 07 June 2024

\* Corresponding author. E-mail: paolo.cozzi@ibba.cnr.it

Preprint submitted at https: //doi.org/10.1101/2024.09.01.610681

## ABSTRACT

Underutilized sheep and goat breeds can adapt to challenging environments due to their genetics. Integrating publicly available genomic datasets with new data will facilitate genetic diversity analyses; however, this process is complicated by data discrepancies, such as outdated assembly versions or different data formats. Here, we present the SMARTER-database, a collection of tools and scripts to standardize genomic data and metadata, mainly from SNP chip arrays on global small ruminant populations, with a focus on reproducibility. SMARTER-database harmonizes genotypes for about 12,000 sheep and 6,000 goats to a uniform coding and assembly version. Users can access the genotype data via File Transfer Protocol and interact with the metadata through a web interface or using their custom scripts, enabling efficient filtering and selection of samples. These tools will empower researchers to focus on the crucial aspects of adaptation and contribute to livestock sustainability, leveraging the rich dataset provided by the SMARTER-database.

**Availability and implementation:** The code is available as open-source software under the MIT license at https://github.com/cnr-ibba/SMARTER-database.

**Subjects** Animal and Plant Sciences, Bioinformatics, Animal Genetics

## STATEMENT OF NEED

### Background

The presence of small ruminant populations is crucial to the socio-economic prosperity of human settlements, particularly in European marginal regions. In these areas, sheep and goat breeds that are not fully utilized have the potential to significantly increase the profitability of small ruminant farming. Their value comes from their distinctive and often unusual genetic composition (e.g., [1, 2]), which helps them adapt to challenging environments, withstand harsh farming conditions, combat biotic and abiotic stressors, and produce high-quality animal-derived products. In this context, the SMARTER (SMAll RuminanTs breeding for Efficiency and Resilience) project [3] developed innovative strategies to improve the resilience and efficiency-related traits of sheep and goats in diverse environments. Here, we present the SMARTER-database, a collection of tools and scripts to gather, standardize, and share with the scientific community a comprehensive dataset of genomic data and metadata information on worldwide small ruminant populations. Existing datasets were scouted from public repositories (see Table 1) and complemented with newly produced data within the context of the SMARTER project [3]. Our system provides a single entry point and standardization tools to explore the genetic diversity and demography of goat and sheep breeds, and to understand the genetic basis of resilience and adaptation, especially in under-utilized breeds.

**Table 1.** Publicly available datasets integrated into the SMARTER database, including chip types, sample sizes, and references. See GigaDB supporting dataset [4] for the full details.

| chip_name | Samples | Reference |
|---|---|---|
| IlluminaOvineHDSNP | 542 | Rochus *et al.* 2018 [5] |
| IlluminaGoatSNP50 | 4653 | Stella *et al.* 2018 [1] |
| IlluminaOvineSNP50 | 2957 | Kijas *et al.* 2012 [2] |
| IlluminaOvineHDSNP | 93 | Rochus *et al.* 2020 [6] |
| IlluminaOvineSNP50 | 1512 | Wang *et al.* 2021 [7] |
| IlluminaOvineHDSNP | 911 | Wang *et al.* 2021 [7] |
| WholeGenomeSequencing | 355 | Wang *et al.* 2021 [7] |
| IlluminaOvineSNP50 | 353 | Beynon *et al.* 2015 [8] |
| IlluminaOvineSNP50 | 116 | Barbato *et al.* 2017 [9] |
| IlluminaOvineSNP50 | 838 | Ciani *et al.* 2020 [10] |
| IlluminaOvineSNP50 | 48 | Belabdi *et al.* 2019 [11] |
| IlluminaOvineSNP50 | 46 | Gaouar *et al.* 2016 [12] |
| IlluminaGoatSNP50 | 364 | Burren *et al.* 2016 [13] |
| IlluminaGoatSNP50 | 523 | Cortellari *et al.* 2021 [14] |

### Data composition

SMARTER-database is mainly composed of two types of data: (i) genotype data, derived from low/high density genome chips and Whole Genome Sequencing (WGS); (ii) phenotype data, including a wide range of information, such as Global Positioning System (GPS) data relative to sampled populations, morphological descriptions of animals, and other production measurements. The objective of our work was to integrate numerous publicly available and newly generated datasets into a single entry point. For genotype datasets, the process of data integration was complicated by the fact that several public datasets were generated years ago and refer to outdated genome assembly versions, using different variant names assigned by different single nucleotide polymorphism (SNP) array



manufacturers, or encoding the same information in different ways. Phenotype data is even more heterogeneous. Missing information is hard to retrieve since animals may no longer be available. Phenotypic data collection often aligns closely with specific experimental objectives, yet descriptions of such data seldom adhere to validated ontological frameworks (e.g., PATO, the Phenotype And Trait Ontology [15]). This lack of standardization complicates cross-study comparisons, hindering robust cross-referencing and comprehensive analytical endeavors. Since the focus of the SMARTER project was on adaptation, a minimal set of attributes was identified for newly collected data: country of origin, breed name, internal IDs used in the genotype file, GPS coordinates, and the main purpose of the breed when available (dairy, meat, wool). However, we encourage the submission of any type of information relevant to resilience, efficiency and adaptation potential.

Since most of the data are derived from SNP chip arrays, the SMARTER-database is structured primarily around this type of information. WGS data, which represents a smaller fraction of the total dataset, is filtered and integrated by aligning the SNPs present in the SNP arrays. While WGS data can provide deeper insights into genetic variation across species, including the ability to perform complex genomic analyses such as pangenome analyses due to recent advances in sequencing technologies [16], bead-chip genotyping remains a highly valuable tool. This is largely because of its cost-effectiveness in analyzing large populations, especially in studies involving livestock or animal breeding, and the fact that SNP arrays yield highly accurate and reproducible genotypes. In contrast, WGS data at low coverage often results in numerous false homozygote calls due to insufficient sequencing depth. In contrast, SNP arrays are specifically designed to minimize such errors, particularly for large-scale genotyping studies [17, 18].

To integrate genotype datasets generated at different times and with different technologies, all files must first be converted to the same format. The majority of data submissions were formatted in PLINK [19], followed by Illumina and Affymetrix formats. Most software packages lack support for proprietary file formats, such as Illumina's row files or Affymetrix's cell data files. Consequently, prior to merging, these proprietary files must undergo conversion into universally accepted formats.

Furthermore, it is imperative to ensure consistency in the reference genome assembly across all datasets. Discrepancies in assembly versions can lead to variations in SNP positions. Even within the same assembly, SNPs may exhibit divergent locations if the mapping procedures between two genotyping array types are not identical [20, 21]. Reliance solely on SNP names was found to be inadequate, given that SNPs from different manufacturers frequently possess unique identifiers and may lack comprehensive information required for unambiguous SNP identification across diverse datasets.

The encoding of genotypes presents another compatibility challenge, as two different formats may be employed to convey identical SNP information. SNPs are identified by aligning an SNP probe (comprising a brief DNA sequence around the SNP) to the reference genome, which may be on either the forward or reverse strand across various genome assemblies. As a result, the same SNPs can be represented by different bases: for example, an [A/G] polymorphism becomes [T/C] if the probe matches on different strands. In such cases, the Food and Agriculture Organization (FAO) Guidelines for Genomic characterization of animal genetic resources [22] recommend reversing the SNP before merging the datasets, and excluding the SNP when the first base is complementary to the second (i.e., [A/T] - [C/G]), since in these cases it is not possible to verify if the SNP is on the same strand on both



**Table 2.** Genotype conversion of DU186191_327.1 (A/G) SNP for four different samples in the SMARTER database. The source version is the assembly version of the received samples, and source coding and genotype are the inferred coding and the received genotype, respectively. The TOP genotype is the final genotype present in the database according to the Illumina TOP [23] convention.

| Smarter_id | Source version | Source coding | Source genotype | TOP genotype |
|---|---|---|---|---|
| UYOA-CRR-000003890 | Oar_v4.0 | Forward | T C | A G |
| UYOA-CRL-000000382 | Oar_v3.1 | A/B | A B | A G |
| NAOA-ADP-000001020 | Oar_v3.1 | Top | G A | G A |
| GROA-CHI-000004137 | Oar_v3.1 | Forward | T T | A A |

**Table 3.** Genotype conversion of OAR1_103790218.1 C/G SNP for four different samples in the SMARTER database. The source version is the assembly version of the received samples, and source coding and genotype are the inferred coding and the received genotype, respectively. The TOP genotype is the final genotype present in the database according to the Illumina TOP [23] convention.

| Smarter_id | Source version | Source coding | Source genotype | TOP genotype |
|---|---|---|---|---|
| UYOA-CRR-000003890 | Oar_v4.0 | Forward | G G | G G |
| UYOA-CRL-000000382 | Oar_v3.1 | A/B | B B | G G |
| NAOA-ADP-000001020 | Oar_v3.1 | Top | G G | G G |
| GROA-CHI-000004137 | Oar_v3.1 | Forward | C C | G G |

assemblies without additional information. Besides, in the A/B format, the letters A and B do not represent the real genotype, but the first and second letters of the SNP recorded in the manifest file. In such cases, it is mandatory to get access to the information in the same manifest file used in the genotyping process. To avoid problems caused by probe alignment and ensure consistent SNP representation regardless of the genome version, the company Illumina has proposed a TOP/BOTTOM format [23], which relies on the SNP probe sequences themselves rather than on probe alignments as in the forward/reverse convention. This solution is not widely adopted: for example, to submit data to EBI-EVA [24], one must provide SNPs in forward orientation with respect to the reference genome (the reference allele needs to match the reference genome in the same position [25]). In 2015, we responded to the need to standardize SNP data [26] by developing SNPchiMp [27] and a series of tools to work with different data sources and to convert data in the same format [28]. In SMARTER, we followed up on the same concept: development of tools to convert genomic data to a reference format (i.e., Illumina TOP) and giving the possibility to convert data according to the user's needs, such as publishing data to public repositories or before merging data with other datasets encoded differently. An example of genotype conversion for *unambiguous* and *ambiguous* SNP is presented in Tables 2 and 3, respectively. During the genotype data import, genotypes are converted to Illumina TOP in order to refer to the same SNP coding across samples from different datasets. It is important to note that after the Illumina TOP conversion, the only available genotypes will be [A/T], [C/G], [A/G], or [A/C], as all possible allele combinations can be converted to one of these four.

## Collecting phenotypes and other metadata

Besides genotype data, other types of data can also be useful when processing and analyzing genotypes in the context of adaptation and genomic selection. Genotype file formats are not suitable for the inclusion of data like geographical coordinates or morphological descriptions of animals. Even though advanced standards like Variant Calling Format (VCF) provide an INFO field supporting user-defined data [29], accessing this kind of information is inefficient because this field cannot be indexed. Preferably, metadata



information should be stored in a dedicated resource where it can easily be queried. In the context of the SMARTER project, a schemaless database was more desirable since metadata do not follow a standard format. A minimal set of requirements should be provided in order to make useful queries; for instance, breed, country, and GPS locations are required in order to retrieve data relevant to adaptation analyses.

## METHODS

### The SMARTER-database project

In order to standardize data and merge genotypes from different datasets, we collected information from SNP chip manufacturers by accessing publicly available manifest files and by directly contacting the manufacturers in case of custom manifest files. Data stored in the SNPchiMp database [27] was also collected as a unique source of information for SNP positions and code conversion. In addition, information from public databases like dbSNP (RRID:SCR_002338) [30] and EVA (RRID:SCR_017425) [31] was integrated in order to provide external reference IDs to the SNPs. To store metadata information, we decided to employ a MongoDB [32] database. This type of database, characterized by its schemaless nature and support for spatial queries, enables the flexible modeling of data with the capability to dynamically add or remove attributes as needed. In addition to sample information, the database is designed to also accommodate variant information, thereby facilitating genotype conversion through reliance on the database. The variant class model defined with the help of the MongoEngine [33] library, a Python-based Object-Document Mapper [34] that facilitates working with MongoDB in an object-oriented way, is presented in Figure 1: a VariantSpecie abstract class defines all the attributes that can be referred to the same SNP, like the different names present in different chip manufacturers or external accession IDs. Multiple Locations are then embedded in the same document in order to track different positions across different assemblies with their specific allele coding. Finally, the VariantSpecie abstract class is extended by the VariantGoat and VariantSheep classes, which serve as the final Object Document Mappers [34] for each sheep and goat variant in the database. All information related to the same SNP is present in the same document, allowing the same SNPs to be tracked even when named differently by different technologies. Moreover, it is possible to identify which SNPs are in common between different datasets and to query data relying on the manufacturer's variant name, Reference SNP cluster IDs, and genomic locations. Finally, the MongoEngine [33] implementation defines some accessory methods that, while not directly represented in MongoDB, can be used to determine the genotype coding or the genotype conversion of a provided SNP according to the assembly information present in the database. More information about the variant model is in the SMARTER-database documentation [35].

Although the VCF [29] file format was proposed as a standard for the distribution of genotype information, we chose to first merge all genotypes we received in a unique PLINK binary file, one for goats and one for sheep. Proprietary file formats (e.g., the Illumina row and the Affymetrix cell files) were converted to PLINK files and merged with other data using the PLINK software (RRID:SCR_001757) [19]. The reason for adopting this format is related to its popularity in the research community: many software applications, libraries, and pipelines focusing on adaptation or genetic diversity rely on this format. In addition, the much smaller size of the PLINK binary file makes it easy to manage (to subset, to index, and more in general to analyze) data using the PLINK software.



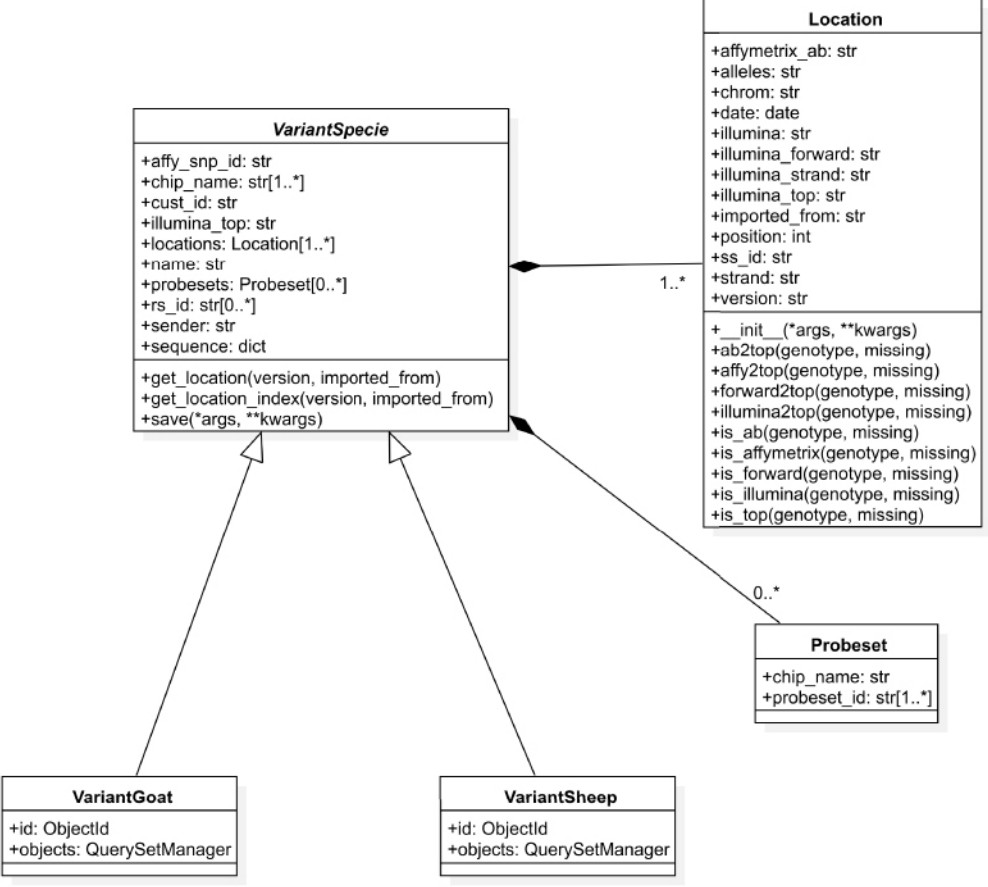

**Figure 1.** A Unified Modeling Language diagram of the Variant classes model implemented in Python: the boxes in the diagram show class attributes and methods, with data types like `str` for string and `dict` for dictionary. The central `VariantSpecie` class serves as an abstract base class, containing common attributes and methods relevant to different species, such as SNP IDs, chip names, and genotype data. The `Location` class models assembly-specific information, including chromosomal positions, alleles, and strand orientation, and is embedded within the `VariantSpecie` class to handle multiple locations per variant. The `VariantSheep` and `VariantGoat` classes inherit from `VariantSpecie`, extending the base class by specializing it for sheep and goat data, respectively. This inheritance allows for the efficient reuse of common functionality while tailoring specific attributes or behaviors for each species. The `Probeset` class, associated with `VariantSpecie`, represents specific data types that come from Affymetrix chips. It captures metadata linked to SNP probes, such as the chip name and probeset ID.

The Illumina TOP/BOTTOM coding convention was selected due to its reliance solely on the probe itself [23]. Its primary advantage lies in ensuring consistent encoding of SNPs across various assembly versions. Consequently, updating SNP positions to accommodate a new reference assembly merely entails a straightforward adjustment.

Different attributes linked to the same SNP, including varying names used by Affymetrix and Illumina or differences in coding types, are leveraged to produce a unified genotype file. This consolidated file guarantees SNP standardization across different datasets by assigning them uniform names and codes, thereby streamlining analyses across samples from various datasets. Furthermore, sample metadata stored in the SMARTER-database helps to identify relevant samples, simplifying their extraction from the comprehensive genotype file relying on the same sample IDs.

The final intersection among all different chip types in sheep will be close to 30K SNPs, including various versions of the Illumina 50K chip (which ranges between 50K and 60K SNPs depending on the release), HD chip, and public and custom Affymetrix chips. This implies that at least 30K SNPs are theoretically shared across all 12K sheep samples; however, all remaining SNPs are reported in the final genotype to support nearly all the 620K sheep SNPs managed by the database. Samples without information on SNPs outside the chip used to generate them will have missing data. The same approach was adopted for goats; however, the number of supported chips is lower, resulting in a final intersection of nearly 93% of the goat-supported SNPs. The data are provided as-is, with only minimal filtering applied based on Identical By State (IBS) to remove duplicate samples when there is overlap between background datasets. This is the only filtering applied to the final dataset; users may apply their own filters after selecting the data they need, avoiding unnecessary filtering for missingness in samples that are irrelevant to their analyses. Standard Minor Allele Frequency (MAF) filters applied as a percentage cannot be applied to the entire dataset, as stated by the FAO guidelines [22], since we would lose all the variability associated with local adaptation, and MAF could change depending on different subsets of samples. Additionally, filtering based on assembly position or sex chromosomes could result in a loss of information: for example, when updating an assembly, unmapped SNPs might be mapped in the new assembly, while previously mapped SNPs could become unplaced. Ideally, a custom remapping of the Illumina and Affymetrix probes against new genome assemblies could increase the intersection between different chip technologies; this can be added in a future release of the database. Sex chromosomes can also be informative, particularly for users interested in reproduction studies, therefore they are not removed. Consequently, we believe that this dataset should be presented without any filtering, leaving it to the user to document all the steps needed to produce the final dataset required for their analyses.

## Reproducibility

The code developed in the SMARTER-database project was enhanced to develop utilities used to keep the database updated, such as adding new data sources, adding new breeds and samples, and converting genotypes in the desired formats in order to produce the final genotype file, as described by our data import guide [36] in the project documentation. The idea behind this implementation is to adhere to the FAIR principles [37] by providing a reproducible and transparent workflow to create and manage the final dataset. Our project is based on the Cookiecutter Data Science Project [38], aiming to standardize data science projects for sharing purposes. It adheres to conventions outlined in the Cookiecutter framework [39], which includes organizing data into specific folders, such as raw data, external data, processing, and final data. Additionally, it includes source folders for importing scripts and libraries, which can be installed as a Python package in a Conda environment (RRID:SCR_018317) [40]. All software dependencies are managed using Conda – a package and environment management system for Python – and tracked through requirements files. Furthermore, it introduces a database folder not found in the original Cookiecutter template. This folder manages database installation and initialization through Docker [41] and docker-compose [42], where Docker is a platform for containerizing applications to ensure they run consistently across different environments. Raw data undergoes initial exploration using IPython (RRID:SCR_001658) notebooks stored in the

**Table 4.** Number of samples with GPS coordinates. Foreground samples are produced within the SMARTER project, while background samples are collected from already published datasets.

|  | Total | With GPS |
|---|---|---|
| Foreground sheep | 5,945 | 79,34% |
| Background sheep | 5,892 | 82,30% |
| Foreground goats | 464 | 87,28% |
| Background goats | 5,540 | 58,34% |

notebook folder, aiming to comprehend its structure and potential issues before importing data into the database. This process includes efforts to infer genotype coding and ensure all necessary information is available for genotype conversion and sample addition to the database.

Subsequently, the dataset is integrated into the database, with updates to available breeds when new ones are added, and the assignment of SMARTER unique and stable IDs to samples, which are used in the resultant genotype files. If metadata is provided, it is appended to the corresponding samples. Currently, the SMARTER database tracks information on approximately 12,000 sheep and 6,000 goats, with nearly 80% and 60% of samples possessing GPS coordinates for sheep and goats, respectively, as indicated in Table 4. In Figures 2 and 3, a graphical representation of samples by country for sheep and goats, respectively, is reported. Figure 4 illustrates the distribution of 12,000 sheep samples based on the genotype technology employed. Despite variations in genotyping technologies, a subset of SNPs are in common: this enables comparison among samples from different datasets. All data processing steps, including environment setup, database initialization, and data processing, are handled using GNU Make [43] in order to provide simple commands to execute all the steps required to produce the final database. These steps, and the importing scripts executed by these steps, are idempotent, meaning that repeating the same command has no side effects, ensuring consistency in the final dataset. This also means that a new dataset can be added simply by updating the Makefile with the new required steps and then calling the proper make command again. The database follows Semantic Versioning [44] to track updates and changes effectively. This project is publicly accessible on GitHub [45], where GitHub workflows automate the process of running Continuous Integration tests that validate critical steps, such as genotype conversion, to ensure that everything functions correctly after each new code contribution.

## Data access

Data can be accessed through two distinct channels: the fully processed PLINK binary genotype files for both goat and sheep are accessible via anonymous FTP, while sample information and metadata are retrievable through SMARTER-backend [46], a RESTful Application Programming Interface (API) interface [47]. Users are required to identify their desired samples using the API interface and then extract their genotypes from the global file. This also means that the resource can be programmatically accessed, thereby enhancing reproducibility and facilitating data integration with other resources.

SMARTER-backend is a Python-Flask application built on top of the SMARTER-database MongoDB, offering a specific URL (i.e., endpoint) to interact with each object stored in the SMARTER-database. Users can query for SNPs, samples, datasets, and breeds by making HTTP requests to the appropriate endpoint and providing the necessary parameters to obtain information in JSON format (Figure 5). Endpoint parameters documentation is



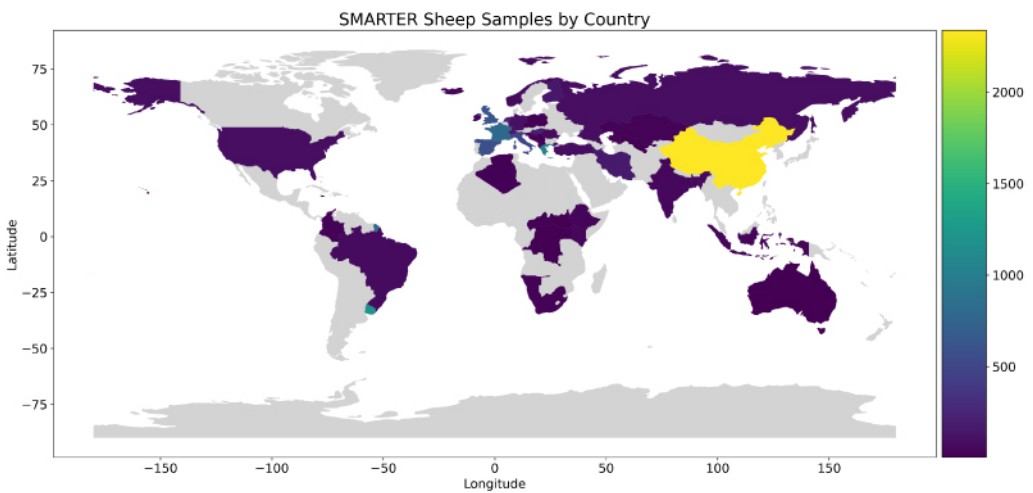

**Figure 2.** Geographical distribution of sheep samples in the SMARTER-database. Countries are colored based on sample count using the Viridis scale, where brighter colors represent higher sample counts.

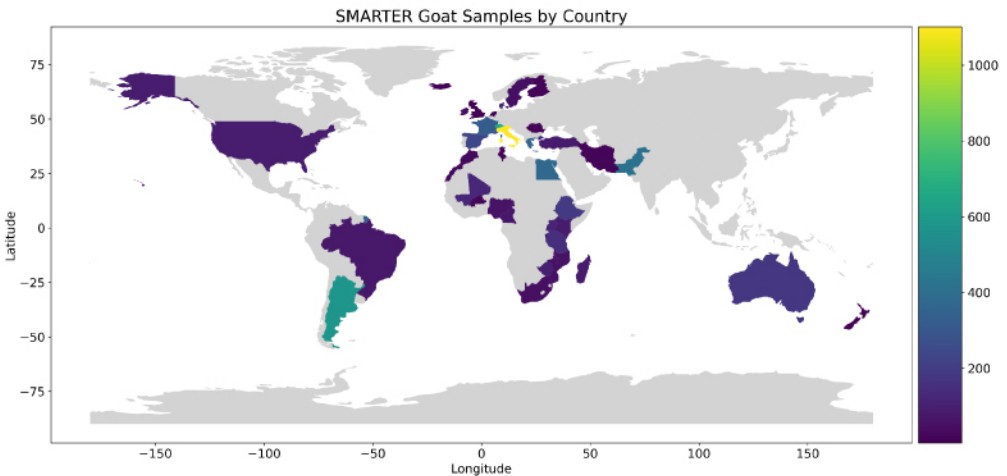

**Figure 3.** Geographical distribution of goat samples in the SMARTER-database. Countries are colored based on sample count using the Viridis scale, where brighter colors represent higher sample counts.

detailed using Swagger [48] and can be accessed via the `/docs` location of the SMARTER-backend itself. Table 5 provides a list of the available endpoints. Figure 6 showcases an example of documentation for the `/sample/goat` endpoint, which enables users to gather information on goat samples. All available parameters are listed along with their descriptions and the supported data types. When the input data type is an array, users can provide the same parameter multiple times. For instance, it is possible to specify multiple breed codes in a single query to obtain all desired samples, as illustrated in Figure 5.

To facilitate data access for partners primarily focused on data analysis with R rather than retrieving data via REST interfaces, we developed the smarterapi R package [49]. This package aims to streamline data retrieval from SMARTER-backend by abstracting the

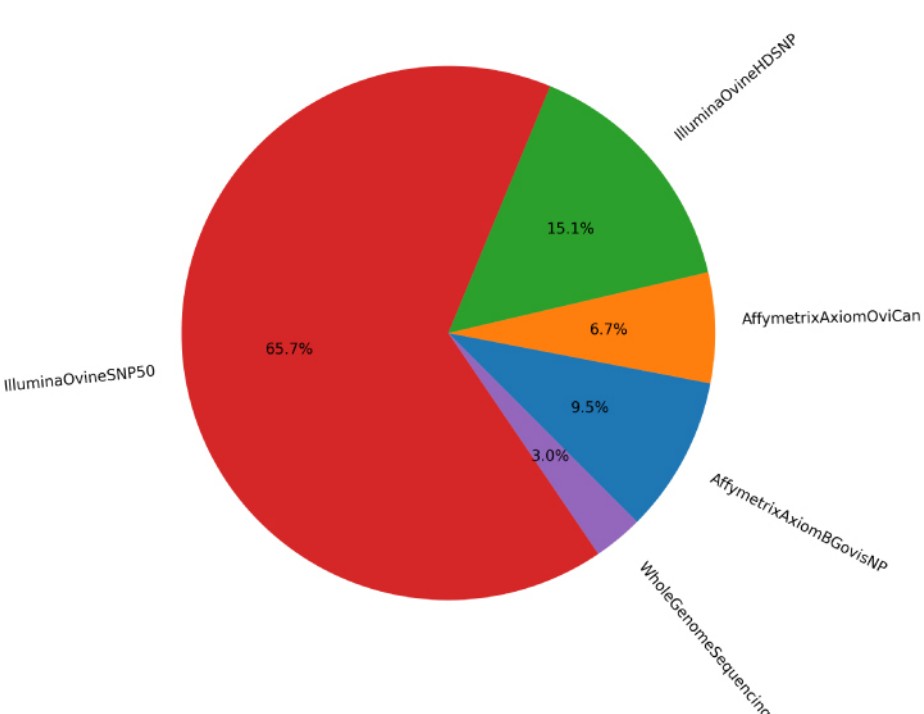

**Figure 4.** Distribution of sheep samples by genotyping technology.

**Figure 5.** Structure of the SMARTER-backend API URL. The endpoint part specifies the desired data type and the parameters to filter out results.

**Table 5.** SMARTER-backend available endpoints. The available data types are described in the SMARTER-database online documentation [35].

| Endpoint suffix | Data type | Description |
| --- | --- | --- |
| /auth/login | Users | User authentication (removed after public release) |
| /breeds | Breed | Available breeds as a list |
| /datasets | Dataset | Available datasets as a list |
| /info | SmarterInfo | Information about database status |
| /samples/sheep | SampleSheep | Sheep samples as a list |
| /samples/goat | SampleGoat | Goat samples as a list |
| /samples.geojson/sheep | GeoJSON | Sheep samples in GeoJSON format |
| /samples.geojson/goat | GeoJSON | Goat samples in GeoJSON format |
| /supported-chips | SupportedChip | Supported genotype platforms as a list |
| /variants/sheep/OAR3 | VariantSheep | Supported sheep SNPs in Oar_v3.1 assembly |
| /variants/sheep/OAR4 | VariantSheep | Supported sheep SNPs in Oar_v4.0 assembly |
| /variants/goat/ARS1 | VariantGoat | Supported goat SNPs in ARS1.2 assembly |
| /variants/goat/CHI1 | VariantGoat | Supported goat SNPs in CHIR_1.0 assembly |

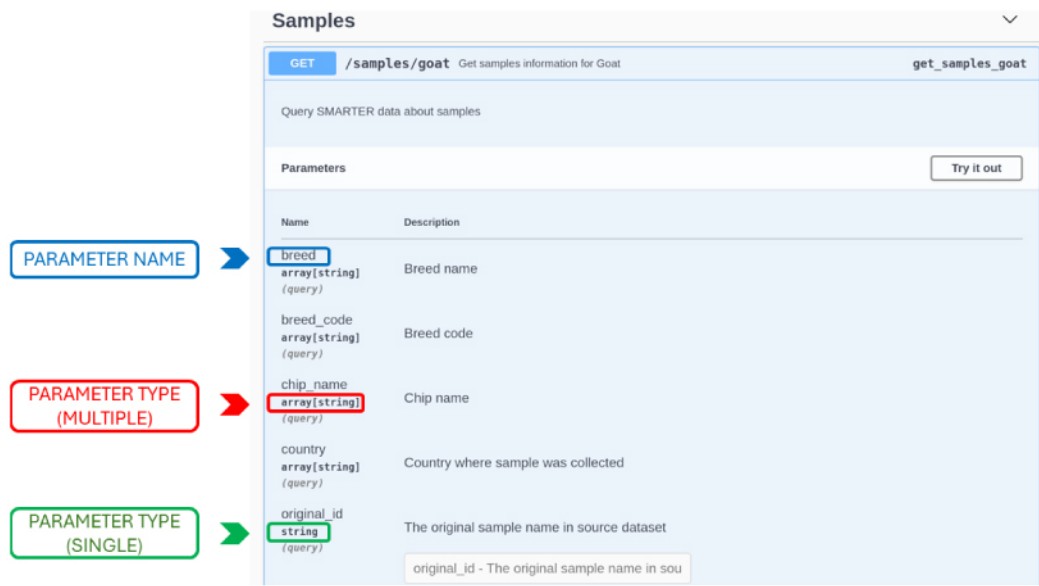

**Figure 6.** Screenshot of the Swagger interface for the SMARTER-backend API. The interface displays available endpoints and parameters, allowing users to interact with and test the API directly. In this example, the GET `/samples/goat` endpoint is shown, with parameters such as breed, breed_code, and chip_name. The `array[string]` type indicates that multiple string values can be entered, enabling filtering by multiple criteria. Users can click the 'Try it out' button to input values, execute queries, and view the results with the URL to generate them, making it easy to explore the API's functionalities.

complexities of handling HTTP requests and result pagination. It offers simple R functions that return data as R data frames, utilizing the same parameters described in the backend documentation. This package facilitates URL generation and data retrieval within R. To assist users, we created an online vignette outlining common operations, such as data retrieval and filtering, as well as more advanced tasks, like working with variants, conducting spatial queries, and extracting data from raster objects using the WorldClim database [50]. These functionalities are valuable for performing landscape genomic analyses on SMARTER data.

The latest tool provided to the community is SMARTER-frontend [51], a web application developed in Angular [52] on top of SMARTER-backend. This application allows users to browse SMARTER data and gain insights into database contents without the need for additional code or software installation. Regardless of the chosen method for collecting samples (API request, R package, or web application), users must identify their samples of interest and extract the required genotypes using PLINK.

## CONCLUSIONS AND FUTURE DEVELOPMENTS

The SMARTER-database project provides valuable information on sheep and goat populations around the world. It is an essential tool for researchers, enabling them to generate new insights and offer the possibility to store new genotypes and drive progress in this field. Comprising a suite of scripts, it standardizes genotype data sourced from various methods and origins, resulting in a unified dataset primed for analyses across different assembly versions for both sheep and goats. Data access is granted to users through the SMARTER-backend, using R packages or web interfaces, while genotypes are available over

FTP. The entire project was developed with the goal of facilitating reproducibility and programmatic data accessibility. The SMARTER database is open to additional dataset integration. Given the schemaless nature of the database and the ability to collect data using the REST API, the most valuable contributions should include metadata, such as GPS coordinates and phenotypes. This type of data will contribute significantly to understanding adaptation and resilience in small ruminants. We plan to support additional assemblies with the option to collect data in VCF format. This will help the community reuse this data and provide the opportunity to upload genotypes to public archives like EBI-EVA. Moreover, by utilizing Illumina TOP coding, users can easily convert data between different assembly versions and project coordinates from older assemblies onto newer ones. This approach ensures accurate SNP positioning across assemblies without concerns about probe orientation. All new improvements are tracked as GitHub issues [53]. When changes are finalized, they are released in a new dataset version, along with any enhancements to data management. All release changes can be viewed in the HISTORY.rst [54] file available on GitHub and in the SMARTER-database Read The Docs documentation [35].

## AVAILABILITY OF SOURCE CODE AND REQUIREMENTS

- Project name: SMARTER-database
- Project home page: https://github.com/cnr-ibba/SMARTER-database
- Documentation: https://smarter-database.readthedocs.io/en/latest/
- Operating system(s): Linux
- Programming language: Python 3.x
- Other requirements: docker, docker-compose, anaconda
- License: MIT
- RRID:SCR_025884

- Project name: SMARTER-backend
- Project home page: https://github.com/cnr-ibba/SMARTER-backend
- Documentation: https://smarter-backend.readthedocs.io/en/latest/
- Operating system(s): Linux
- Programming language: Python 3.x
- Other requirements: docker, docker-compose
- License: GPL-3.0

- Project name: smarterapi
- Project home page: https://github.com/cnr-ibba/r-smarter-api
- Documentation: https://cnr-ibba.github.io/r-smarter-api/
- Operating system(s): Platform independent
- Programming language: R
- Other requirements: gdal
- License: GPL-3.0

- Project name: SMARTER-frontend
- Project home page: https://github.com/cnr-ibba/SMARTER-frontend
- Operating system(s): Linux
- Programming language: TypeScript
- Other requirements: NodeJS, npm, angular
- License: MIT

## DATA AVAILABILITY

Supporting data are available from the GigaDB repository [4]. The genotype files supporting the results of this article are also available via anonymous FTP at ftp://webserver.ibba.cnr.it or through the *smarterapi* R package (see https://cnr-ibba.github.io/r-smarter-api/articles/smarterapi.html#collect-genotypes). Metadata and sample information can be accessed via the SMARTER-backend REST API at https://webserver.ibba.cnr.it/smarter-api/ using the *smarterapi* R package (https://cnr-ibba.github.io/r-smarter-api/) and through the database portal at https://webserver.ibba.cnr.it/smarter/.

## ABBREVIATIONS

API, Application Programming Interface; FAO, Food and Agriculture Organization; FTP, File Transfer Protocol; GPS, Global Positioning System; MAF, Minor Allele Frequency; SMARTER, SMAll RuminanTs breeding for Efficiency and Resilience; SNP, single nucleotide polymorphisms; VCF, Variant Calling Format; WGS, Whole Genome Sequencing.

## DECLARATIONS

### Ethical approval

Not applicable.

### Competing interests

The author(s) declare that they have no competing interests.

### Authors' contributions

AS, BS, GC, and AT conceptualized the study. VT, PP, AMJ, JJA, FF, SK, GTK, GC, AT, BS, and AS provided resources. Methodology was developed by PC, BS, AS, AT, and GTK. PC and KG developed the software. Data curation was performed by PC, VT, KG, PP, AMJ, JJA, FF, SK, and GTK. Formal analysis was conducted by AM, JRD, VT, FB, and LP. PC and AS prepared the original draft. AM, JRD, BS, FB, and RR reviewed and edited the manuscript. PC was responsible for visualization. AS and BS supervised the project. AS and BS acquired funding. RR managed the project.

### Funding

This project has received funding from the European Union's Horizon 2020 research and innovation program under grant agreement No. 772787 (SMARTER).

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
