## [Editor Report]

Editor’s AssessmentThis paper presents the SMARTER database, a collection of tools and scripts to gather, standardize, and share with the scientific community a comprehensive dataset of genomic data and metadata information on worldwide small ruminant populations. Which has come out of the EU multi-actor (12 country) H2020 project called SMARTER: SMAll RuminanTs breeding for Efficiency and Resilience. This bringing together genotypes for about 12,000 sheep and 6,000 goats, alongside phenotypic and geographic information. The paper providing insight into how the database was put together, presenting the code for the SMARTER—frontend, backend and API, alongside instructions for users. Peer review tested the platform and provided suggestions on improving the metadata. Demonstrating the project provides valuable information on sheep and goat populations around the world, that can be an essential tool for ruminant researchers. Enabling them to generate new insights and offer the possibility to store new genotypes and drive progress in the field.Editor’s AssessmentThis paper presents the SMARTER database, a collection of tools and scripts to gather, standardize, and share with the scientific community a comprehensive dataset of genomic data and metadata information on worldwide small ruminant populations. Which has come out of the EU multi-actor (12 country) H2020 project called SMARTER: SMAll RuminanTs breeding for Efficiency and Resilience. This bringing together genotypes for about 12,000 sheep and 6,000 goats, alongside phenotypic and geographic information. The paper providing insight into how the database was put together, presenting the code for the SMARTER—frontend, backend and API, alongside instructions for users. Peer review tested the platform and provided suggestions on improving the metadata. Demonstrating the project provides valuable information on sheep and goat populations around the world, that can be an essential tool for ruminant researchers. Enabling them to generate new insights and offer the possibility to store new genotypes and drive progress in the field.

---

## [Reviewer Report]

Indicate in the comments box below whether you are happy with the changes made or if the manuscript is unacceptable.Comments on revised manuscriptMy comments have been properly addressed. The manuscript is acceptable for publication.

---

## [Reviewer Report]

Indicate in the comments box below whether you are happy with the changes made or if the manuscript is unacceptable.Comments on revised manuscriptThe revision has improved considerably the manuscript. I understand that the use of several terms that are better understandable to information scientist than to molecular biologists is unavoidable and most terms have been briefly explained. However, there are still a few issues (numbering of my previous review). 1. The title has not been changed and still is misleading by suggesting the use WGS data. Even if the abstract is more informative, a title rajsing expectations that are not fulfilled may irritate the readers. Suggested title: SMARTER-database: a tool to integrate SNP array datasets for sheep and goat breeds. 2. I am afraid that the authors have misunderstood my point. Many readers may wonder why genome-wide arrays are still useful if much more powerful WGS datasets are now commonly used. Therefore, it may just be mentioned that SNP arrays offer allow an affordable analysis of many animals and yield for a given panel of genotypes much fewer false homozygote than normal-coverage WGS data. That’s all and it is not outside the scope of this paper. I understand that this paper is part of a project aimed at local adaptation, but readers will appreciate a versatile general-purpose database rather than a prelude to future publications of this project. 3. OK. 4. As in any scientific paper, the use of previous data should be transparent and be supported by a complete bibliography. Because this can be realized by a simple Excel sheet, there should be no need to use any script for this. However, the supplemental table is disorganized and should be more complete: a. Please order the entries according to the species. b. Readers may want to quickly look if there are data on a specific breed, but then “27 pops” etc. is less than helpful, especially if the source of the data has not been mentioned. So please mention all breeds separately. We also recommend to give breeds all a separate line (this gives >500 lines, but this is still surveyable; you may also do this in a separate table) and give the number of individuals per breed in column F, thus enable users to quickly scroll through the data before using the REST API tool. c. Please specify the abbreviations used in columns G and K d. It is a good idea to mention the doi for a quick access to the source literature, but these appear not to be available for several sources. Please specify the source as done normally, e.g., “Kijas et al., 2023”; or “<source lab>, unpublished”. e. I guess that the processed source datasets only contain unique data, i.e., without including breeds that for a given source have been copied from previous sources (often from Kijas et al., 2012). This should be mentioned. REST API seems to be an excellent, useful and easy tool, but propagating its use does not constitute a scientific advance that normally is the primary goal of a scientific paper. 5. For the moment, OK, although general-purpose databases should aim to be comprehensive. You may consider a list of publications on sheep and goat SNP data that have not yet been integrated into SMARTER. 6. OK, clear. 7. OK. 8. OK. 9. OK. 10. OK. 11. (cf. point 2 raised by the other reviewers.) Oar_v3.1 is indeed the standard assembly used in papers reporting sheep HD sheep HD datasets. However, I would not miss the Oar_v3.1 option, Instead, including the option to output the ARS_UI_Ramb_v3.0 would facilitate the use of an assembly with superior contiguity and quality and thus improve the results of the downstream analysis. 12. Most users are not aware of the possible errors in published datasets (duplicates, mislabeling, outliers) and it would be useful to stress the need for a critical inspection of data . 13. OK. I recommend 1-ibs NJ trees of PCA for checking on outliers because it differentiates between mislabelling (outliers completely outside the breed cluster) and crossbreeding (outlier still near breed cluster, but attached close to the root). PCA may also miss duplicates. 14. OK 15. OK. 16. OK. I appreciate the extensive explanation, but it is still not entirely clear to me what is the relation between the output of the Swagger interface and the output of REST API (or do I not ask a sensible question?). 17. As for 16. 18. OK. 19. OK. 20. OK. 21. OK.

---

## [Reviewer Report]

Reviewer name and names of any other individual's who aided in reviewerRan LiDo you understand and agree to our policy of having open and named reviews, and having your review included with the published manuscript. (If no, please inform the editor that you cannot review this manuscript.)YesIs the language of sufficient quality?YesPlease add additional comments on language quality to clarify if neededIs there a clear statement of need explaining what problems the software is designed to solve and who the target audience is? YesAdditional CommentsIs the source code available, and has an appropriate Open Source Initiative license <a href="https://opensource.org/licenses" target="_blank">(https://opensource.org/licenses)</a> been assigned to the code?YesAdditional CommentsAs Open Source Software are there guidelines on how to contribute, report issues or seek support on the code?YesAdditional CommentsIs the code executable?YesAdditional CommentsIs installation/deployment sufficiently outlined in the paper and documentation, and does it proceed as outlined?YesAdditional CommentsIs the documentation provided clear and user friendly?YesAdditional CommentsIs there enough clear information in the documentation to install, run and test this tool, including information on where to seek help if required?YesAdditional CommentsIs there a clearly-stated list of dependencies, and is the core functionality of the software documented to a satisfactory level?YesAdditional CommentsHave any claims of performance been sufficiently tested and compared to other commonly-used packages? YesAdditional CommentsIs test data available, either included with the submission or openly available via cited third party sources (e.g. accession numbers, data DOIs)?Additional CommentsAre there (ideally real world) examples demonstrating use of the software? YesAdditional CommentsIs automated testing used or are there manual steps described so that the functionality of the software can be verified?Additional CommentsAny Additional Overall Comments to the AuthorThe authors presented an online SMARTER-database, which collected a large number of genotype data for sheep and goats. The resources are of great importance for the community. My major concerns: 1) The below link is not accessible: webserver.ibba.cnr.it 2) For sheep, the database support reference genome assembly of Oar3 and Oar4, but actually Oar 3 is rarely used. Instead, the current ovine reference genome assembly (ARS-UI_Ramb_v3.0) is not supported. 3) For the presentation of metadata (https://webserver.ibba.cnr.it/smarter/breeds?species=Sheep), I suggest additional columns indicating the region and country should be provided. 4) For the datasets (https://webserver.ibba.cnr.it/smarter/datasets), references are needed to know where the data are from.RecommendationMajor Revisions

---

## [Reviewer Report]

Reviewer name and names of any other individual's who aided in reviewerJohannes A. LenstraDo you understand and agree to our policy of having open and named reviews, and having your review included with the published manuscript. (If no, please inform the editor that you cannot review this manuscript.)YesIs the language of sufficient quality?YesPlease add additional comments on language quality to clarify if neededIs there a clear statement of need explaining what problems the software is designed to solve and who the target audience is? YesAdditional CommentsThis is implicitly clear and does not need to elaborate upon.Is the source code available, and has an appropriate Open Source Initiative license <a href="https://opensource.org/licenses" target="_blank">(https://opensource.org/licenses)</a> been assigned to the code?YesAdditional CommentsAs Open Source Software are there guidelines on how to contribute, report issues or seek support on the code?NoAdditional CommentsThis does not to seem necessaryIs the code executable?Unable to testAdditional CommentsIs installation/deployment sufficiently outlined in the paper and documentation, and does it proceed as outlined?Unable to testAdditional CommentsIs the documentation provided clear and user friendly?YesAdditional CommentsI did not test thisIs there enough clear information in the documentation to install, run and test this tool, including information on where to seek help if required?YesAdditional CommentsI did not test thisIs there a clearly-stated list of dependencies, and is the core functionality of the software documented to a satisfactory level?NoAdditional CommentsI did not see such a list, but I would not be able to assess thisHave any claims of performance been sufficiently tested and compared to other commonly-used packages? Not applicableAdditional CommentsIs test data available, either included with the submission or openly available via cited third party sources (e.g. accession numbers, data DOIs)?YesAdditional CommentsAre there (ideally real world) examples demonstrating use of the software? YesAdditional CommentsIs automated testing used or are there manual steps described so that the functionality of the software can be verified?NoAdditional CommentsI did not find any of this but it does not seem to be essential.Any Additional Overall Comments to the AuthorThis manuscript describes a highly useful database of sheep and goat genome-wide SNP genotypes from several sources, supplemented with phenotypes and geographic locations. I recommend this manuscript for publication in Gigascience after a revision. There is some missing information, whereas the presentation should become less cryptic to readers who are less familiar with the bioinformatic terminology. Missing info. 1. The title and abstract do not mention that SMARTER focuses on SNPs that are genotyped on bead arrays or related technologies. The focus on the genome-wide (GW) SNP genotypes, which only partially represents the total genomic diversity, should already be clear from the Title and the Abstract. 2. Nowadays there are more publications on WGS data, T2T sequences and pangenomes than on GW SNP genotypes, so people may wonder if the GW SNP genotypes still are useful. It may be emphasized that bead-arrays allow an affordable analysis of many animals and that genotypes derived from WGS data contain many false homozygote scores if not sequenced at a very high coverage. 3. Figures 2 and 3 give an idea of the geographic coverage, but what is the unit of the numbers that are visualized in the heat map (0 to 2300 for sheep, 0 to 1100 for goats)? 4. It is not clear which published data have been used or not. We recommend presenting a supplemental table describing the current contents: country, breed, number of animals, number of SNPs (at least 50K or HD), reference. 5. Is there an organized effort to update the database, which ideally should contain all published GW SNP databases? 6. To my experience for most HW SNP datasets only the filtered data after quality control (typically 45 to 49K, less than 42K if sheep 50K and HD genotypes are combined) are available. How is this handled? 7. It may be mentioned that after omission of A/T and G/C SNPs the TOP strand consists only of A/C and A/G SNPs. 8. The problematic SNPs are mentioned twice within the last paragraph of the section Data Composition. 9. Does SMARTER allow to store phased datasets and show the variant haplotypes? These can now be generated by long-read sequencing and are needed for several downstream analysis options. 10. Table 1: OAR3 = Oar_v3.1 and OAR4 = Oar_v4.0? Please use the official codes. 11. Are there options to convert the data to newer assemblies? For instance, the sheep ARS-UI_Ramb_v3.0 is superior to Oar_v4.0. I have used an NCBI tool for conversion of Oar_v1.0 (most popular for 50K datasets) and Oar_3.1 (used often for sheep HD datasets) to Oar_v4.0, but this tool has probably been discontinued and was not available for goat assemblies. 12. I repeatedly found that most published or unpublished databases contain several errors such as duplicates and outliers by mislabeling or crossbreeding. Because these are better removed prior to downstream analysis, data curation would be desirable, for instance by an inspection of a NJ tree of individuals. This also shows the degree of breed-level differentiation, for instance the relationships of different populations of a transboundary breed. These caveats should at least be mentioned. 13. Another caveat: is there a systematic check on the validity of the merging of datasets by testing if breeds sampled independently by different institutes cluster closely together? Presentation. 14. Abbreviations should not be used in abstract. What is “REST API”? These abbreviations of course are in the list, but what is “Representational State Transfer”? And “JSON Web Token”? 15. Figure 1 needs more guidance via the legend. The boxes show alternative formats? What are “str”, “dict “? 16. Figure 5 is useful and seems to retrieve data for the goat Alpine and Bionda dell'Adamello breeds. It would also be useful to show other “API-URL” (this is user input?) while describing in plain language what is being accomplished. 17. Figure 6: bold indicates the user input? What is exactly a “array [string]” (give an example)? A few other examples may be most instructive and familiarize the reader with the logic of SMARTER. 18. In the section “The SMARTER-database project”: what is a mongoengine? 19. In the same section: “Finally the VariantSpecie abstract class is inherited by . . .”: this sentence is difficult to understand. 20. In the section Reproducibility: please give a short description of what is the use of the Conda and Docker programs. 21. Same section: “Raw data undergoes initial exploration”, “structure and potential issues”: can you be more specific? The last part of this section is also difficult to follow.RecommendationMajor Revisions